# High numbers of differentiated CD28null CD8+ T cells are associated with a lowered risk for late rejection and graft loss after kidney transplantation

**Michiel G. H. Betjes**⬤*, **Nicolle H. R. Litjens**

Department of Internal Medicine, section Nephrology and Transplantation, Erasmus MC, University Medical Centre, Rotterdam, the Netherlands

* m.g.h.betjes@erasmusmc.nl

**Data Availability Statement:** All relevant data are within the manuscript and its Supporting Information files

## Abstract

### Background

The hypothesis was tested that parameters of an aged T-cell compartment associate with the risk for late rejection after kidney transplantation.

### Methods

Recipients of a kidney transplant in the period 2007–2013 were (N = 365) were included. T cells were characterized prior to transplantation by flow cytometry as naive (CD45RO⁻CCR7⁺), central-memory (CD45RO⁺CCR7⁺), effector-memory (CD45RO⁻CCR7⁻) or terminally differentiated CD8⁺ Temra (CD45RO⁻/CCR7⁻/CD28⁻) cells. T cell telomere length and thymic output were assessed prior to transplantation in 202 recipients. Follow-up was until December 2018. The date of the first time of biopsy-proven late rejection (>6 months after transplantation) was used to calculate the rejection-free survival time.

### Results

Fifty cases of biopsy-proven rejection were recorded. Thymic output and T cell telomere length did not associate with late rejection-free survival. However, the percentage and absolute numbers of CD8⁺Temra and CD28null CD8⁺ T cells were significantly lower in patients with late rejection. Specifically, in the highest tertile of percentages of CD28null CD8⁺ T cells, the cumulative incidence of late rejection at 5 and 10 years was only 5% and 8% compared to 16% and 20% in the middle to lowest tertile (p = 0.002). Multivariate proportional hazard analysis showed that percentage and absolute number of CD28null CD8⁺ T cells remained significantly associated with late rejection and rejection-related graft loss.

### Conclusion

High numbers of differentiated CD28null CD8⁺ T cells decrease the risk for late rejection and rejection-related graft loss after kidney transplantation.

**Funding:** This study was supported by a grant (KSPB.10.12) of the Dutch Kidney Foundation (http://www.nierstichting.nl/) awarded to Dr Betjes. The funders had no role in study design, data collection and analysis, decision to publish, or preparation of the manuscript.

**Competing interests:** The authors have declared that no competing interests exist.

# Introduction

Progressive loss of renal function leading to end-stage renal disease (ESRD) is associated with premature ageing of the T-cell system. The pro-inflammatory environment resulting from loss of renal function results in a lower thymic output, increased T-cell differentiation, telomere shortening and skewing of the T cell receptor (TCR) repertoire[1–7].

The changes in the peripheral T-cell compartment of ESRD patients resemble the physiological changes in the immune system of healthy elderly individuals with the immunological T-cell age of ESRD patients on average being increased by 15–20 years compared to their chronological age[1].

Increasing age of the recipient decreases the risk for acute rejection after kidney transplantation, which is believed to be a result of an age-related less alloreactive immune system[8, 9]. We have investigated the relation between several immunological parameters of ageing (thymic output, T cell telomere length and T cell differentiation status) and early rejection of the kidney allograft. Only T cell differentiation status appeared to be associated with acute rejection after kidney transplantation[10–12]. Unexpectedly, in particular the presence of large numbers (both in percentage as absolute number) of T cells without surface expression of CD28 was associated with a decreased risk for allograft rejection[10, 12]. The CD28 molecule is a pivotal part of an important co-stimulatory pathway involving interaction with CD80/CD86 on antigen-presenting cells[13, 14].

The CD28null T cells are mainly present in the circulating memory T cell compartment and relatively few are found in the lymph nodes[15]. Their presence within the classical T cell subsets increases with progressive T cell differentiation. For example, all naïve T cell are CD28 positive while the vast majority of highly differentiated Temra cells has lost CD28 expression on their cell surface. As a consequence, CD28null T cells need exogenous cytokine signals like IL-15 and IL-21 to become activated and start proliferation after T cell receptor-allogeneic HLA interaction[16–18].

Currently, it is recognized that late rejection is a major cause of allograft loss in the long-term and the adequate treatment of this condition is considered an unmet need in the field of kidney transplantation[19–21]. Whether the status of the immune system of the recipient is a significant risk factor for late allograft rejection is currently unknown. To address this question we analyzed the status of the T cell system prior to transplantation, in particular the presence of CD28null T cells, with the incidence of late rejection.

# Patients and methods

## Patients

For this study we combined the datasets of 2 previous studies with at least 6 months follow-up after transplantation, including recipients (n = 365) of a kidney transplant within the period 2007–2013 at our transplantation center. The first dataset included consecutive recipients (n = 158) of a kidney transplant in the period 1 January 2007 to 1 December 2009 of whom the peripheral T cells were immunophenotyped before transplantation[10]. This study was approved by the Medical Ethical Committee of the Erasmus MC (MEC-2007-228). The second dataset included recipients (n = 207) that participated in a randomized-controlled clinical trial with the primary aim to study the efficacy of a genotype-based approach to tacrolimus dosing [22]. All patients undergoing a living-donor kidney transplantation (KT) in the period from 1 November 2010 to 1 October 2013 were considered for participation in this study. Donor kidneys were procured by Eurotransplant in case of deceased donors kidneys and the Erasmus MC for living donor kidney transplantations.

The trial and sub study involving the immunophenotyping of peripheral blood T cells was approved by the Medical Ethical Committee of the Erasmus MC (MEC-2010–080). All patients

gave written informed consent to participate in the studies and it was conducted in accordance with the Declaration of Helsinki and the Declaration of Istanbul. Patients were excluded if they were younger than 18 years and if they received immunosuppressive medication (except for glucocorticoids) within 28 days prior to transplantation. None of the transplant donors was from a vulnerable population and all donors or next of kin provided written informed consent that was freely given. Consent for donation and donor registration in case of postmortem donation was obtained and regulate as required by national legislation within the Eurotransplant region (www.eurotransplant.org under legislation/eurotransplant). In our cohort, sixty percent of deceased donor kidneys were from deceased by cardiac death donors and 40% from deceased by brain death donors. The informed consent form for living kidney donors (translated in English) is shown in S1 Fig.

All medical costs were covered by the medical insurance.

Initial maintenance immunosuppression with tacrolimus, mycophenolate mofetil and glucocorticoids was given in >90% of patients (Table 1 As per protocol the following dosages and through levels were used: tacrolimus aiming for predose concentrations of 10–15 ng/mL in weeks 1–2, 8–12 ng/mL in weeks 3–4, and 5–10 ng/mL, thereafter, mycophenolate mofetil starting dose of 1 g b.i.d., aiming for predose concentrations of 1.5–3.0 mg/L. All patients received 50 mg prednisolone b.i.d. intravenously on days 0–3. Thereafter, 20 mg oral prednisolone was started and subsequently tapered to 5 mg at month 3.

The HLA-typing was assessed according to the international standards (American Society for Histocompatibility and Immunogenetics/the European Federation for Immunogenetics) using serologic and DNA-based techniques. The panel reactive antibodies (PRA) were determined at the laboratory of the blood bank in Leiden, the Netherlands.

All transplantations were ABO-compatible with a negative complement dependent cross match. Flowcytometry based cross matches were not performed and the presence of donor-specific anti-HLA antibodies by Luminex was not regularly assessed.

Late rejection was defined as biopsy-proven allograft rejection diagnosed at least 6 months after KT using the Banff criteria 2015. Graft loss was defined as the need for dialysis or retransplantation. The last day of follow-up was December 1, 2018.

## PBMC isolation

By using Ficoll-Paque Plus (GE healthcare, Uppsala, Sweden), peripheral blood mononuclear cells (PBMC) were isolated from heparinized blood samples. Blood was drawn from KT-recipients the day before KT. Isolated PBMCs were stored at -150˚C with a minimum amount of $10 \times 10^6$ cells per vial until further analysis.

## T cell differentiation status and absolute numbers of T cell subsets

To determine the T-cell differentiation status, staining for CD4 and CD8 was combined with the differential expression of CCR7 and CD45RA to identify naïve and memory T cells as has been described before in detail. Memory T cells were further subdivided into central-memory T cells, effector-memory T cells and the most differentiated Temra cells[23, 24]. In addition, CD28 expression was measured on T cells, which allows for a clear distinction between CD28 positive and negative T cells[25, 26].

## Relative telomere length (RTL) and recent thymic emigrants

Only for the recipients of a kidney transplant after 2010 (n = 202), the RTL and number of recent thymic emigrants (RTE) were assessed. To determine the RTL of CD4$^+$ and CD8$^+$ T

**Table 1. Clinical and demographical characteristics of kidney transplant recipients prior to transplantation.**

| | No late rejection (n = 315) | Late rejection (n = 50) | p-value |
|---|---|---|---|
| Age in years, median (range) | 55 (18–79) | 51 (19–78) | 0.004 |
| Male/female | 205/110 | 31/18 | 0.9 |
| Follow-up time in months, median (IQR) | 82 (7–137) | 71 (8–137) | 0.03 |
| Time to diagnosis of late rejection, months median (range) | - | 44 (7–112) | |
| Living kidney donor (% within group) | 88.6% | 91.8% | 0.5 |
| Previous kidney transplant | 9.9% | 24.5% | 0.02 |
| Pre-emptive transplantation | 48.7% | 20.4% | 0.007 |
| Early rejection (within first 6 months after KT) | 17.5% | 32.7% | 0.02 |
| PRA at time of transplantation (average with SD) | 2.9% (10.3) | 5.4% (17.8) | 0.3 |
| PRA peak serum (average with SD) | 8.8% (19.4) | 16.4% (17.7) | 0.07 |
| Total number of HLA mismatches (mean) | 3.6 | 3.4 | 0.4 |
| Induction therapy: | | | 0.2 |
| • Basiliximab | 59.4% | 44.9% | |
| • ATG | 2.9% | 4.1% | |
| Maintanance immune suppression | | | 0.2 |
| • tacrolimus/mycophenolate/prednisone | 95.5% | 91.8 | |
| • other | 4.5% | 8.2% | |
| Distribution of underlying kidney disease | | | 0.8 |
| • Nephrosclerosis/hypertension | 27.0% | 26.5% | |
| • Primary glomerulopathies | 19.4% | 18.74% | |
| • Diabetes mellitus | 17.1% | 10.2% | |
| • Urinary tract infections/ stones | 1.6% | 2.0% | |
| • Reflux nephropathy | 5.4% | 8.2% | |
| • Polycystic kidney disease | 15.6% | 14.3% | |
| • Other | 8.6% | 14.3% | |
| • Unknown | 5.4% | 6.1% | |
| CMV seropositive* | 65.3% | 61.2% | 0.6 |
| Type of late rejection | | | |
| • ABMR | | 39 | |
| • TCMR | | 5 | |
| • Mixed rejection | | 6 | |

*CMV seropositive: detectable serum antibodies against cytomegalovirus at time of transplantation

cells, flow fluorescent in situ hybridization was performed as described in detail previously[1]. RTE were defined as CD31-expressing naïve T cells[27].

## Statistics

The primary outcome of this study was the incidence of late rejection after kidney transplantation.

The difference between continuous variables was assessed with the Mann–Whitney U test. The difference between categorical variables was analyzed either with the Pearson's chi-squared test or with the Fisher's exact test depending on the expected values in any of the cells of a contingency table. Univariate and multivariate Cox proportional hazard analysis was used to assess the association between immunological and clinical parameters, and the outcome late biopsy proven rejection. Clinical and immunological parameters that had a p-value of <0.1 in the univariate analysis were tested in the multivariate Cox regression analysis using forward

and backward modeling. Kaplan-Meier survival curves were made for late rejection and graft loss, dividing the patients into groups according to the tertile of either percentage or absolute number of CD8CD28null T cells before transplantation.

The significance level (p-value) was two-tailed and an α of 0.05 was used for all analyses. Statistical analyses were performed using SPSS® version 21.0 for Windows® (SPSS Inc., IL, USA) and GraphPad Prism 5 (CA, USA). Figures were created with GraphPad Prism 5 (CA, USA).

## Results

### Recipients characteristics and late allograft rejection

Recipients characteristics are shown in Table 1. Fifty recipients (median age 51 year, range 19–78) were diagnosed with a late rejection at a median of 44 months after transplantation (range 7–112 months). The group of recipients with late rejection differed from the no late rejection group with respect to age at transplantation (younger), a previous kidney allograft (more frequently), an early acute rejection (more frequently) and the frequency of renal replacement therapy before kidney transplantation (more often). Twenty recipients lost their allograft because of late rejection.

### Pre-transplant T cell ageing parameters and late allograft rejection

The average distribution of different T cell subsets given as percentages of the total of circulating CD4 or CD8 T cells, is shown in Table 2. The recipient group with late rejection had a significantly higher percentages of naïve CD4 and CD8 T cells, and lower percentages of the CD8 effector-memory and terminally differentiated Temra cells. The latter finding corresponded with a significant lower percentage of CD28null CD8 T cells. Of note, the relative telomere length and number of CD31 positive naïve T cells (reflecting thymus output of T cells) was not significantly different between both groups (Table 2).

The absolute numbers of different T cells subsets (Table 3) revealed that the significant shifts in CD8 T cell subsets in the late rejection group could be attributed to an expansion of highly differentiated T cells as reflected by higher numbers of Temra cells and CD28null CD8 T cells. Average absolute numbers of naïve T cells were similar between the no late rejection and late rejection groups, underlying the importance of absolute cell counts for correct interpretation of relative shifts in T cell subsets.

**Table 2. Characteristics of circulating T cells with subsets given in percentages prior to transplantation in relation to biopsy-proven late kidney rejection.**

|  | no late rejection (n = 315) | late rejection (n = 50) | p-value |
|---|---|---|---|
| Naive CD4 T cells % | 30.6 ± 0.9 | 36.0 ± 2.4 | 0.03 |
| CD31pos naïve CD4 T cells* % | 64.5 ± 1.1 | 67.6 ± 3.8 | 0.39 |
| Central memory CD4 T cells % | 37.7 ± 0.8 | 33.7 ± 1.9 | 0.07 |
| Effector memory CD4 T cells % | 25.1 ± 0.8 | 21.6 ± 2.0 | 0.10 |
| CD28null CD4 T cells % | 6.4 ± 0.6 | 4.6 ± 1.0 | 0.23 |
| Naïve CD8 T cells% | 22.7 ± 1.1 | 30.9 ± 2.8 | 0.005 |
| CD31pos naïve CD8 T cells* % | 95.4 ± 0.5 | 95.0 ± 1.2 | 0.78 |
| Central memory CD8 T cells % | 7.2 ± 0.3 | 8.1 ± 1.0 | 0.31 |
| Effector memory CD8 T cells | 29.3 ± 0.9 | 24.3 ± 2.3 | 0.04 |
| Temra CD8% | 30.8 ± 1.1 | 24.5 ± 2.3 | 0.02 |
| CD28null CD8 T cells% | 41.6 ± 1.3 | 32.5 ± 2.7 | 0.003 |
| Relative telomere length CD4 T cells* | 12.7 ± 0.4 | 12.5 ± 0.4 | 0.93 |
| Relative telomere length CD8 T cells* | 12.5 ± 0.9 | 11.7 ± 1.0 | 0.44 |

**Table 3. Circulating numbers of T cell subsets in cells/μl prior to transplantation in relation to late kidney rejection.**

|  | no late rejection (n = 315) | late rejection (n = 50) | p-value |
|---|---|---|---|
| Naive CD4 T cells | 142 ± 11 | 102 ± 23 | 0.16 |
| Central memory CD4 T cells | 240 ± 12 | 206 ± 20 | 0.15 |
| Effector memory CD4 T cells | 149 ± 8 | 120 ± 11 | 0.17 |
| CD28null CD4 T cells | 40 ± 4 | 31 ± 7 | 0.34 |
| Naïve CD8 T cells | 51 ± 5 | 50 ± 12 | 0.92 |
| Central memory CD8 T cells | 83 ± 7 | 116 ± 9 | 0.92 |
| Effector memory CD8 T cells | 25 ± 2 | 25 ± 3 | 0.13 |
| Temra CD8 | 145 ± 10 | 85 ± 11 | 0.0001 |
| CD28null CD8 T cells | 179 ± 13 | 104 ± 16 | 0.0003 |

## Multivariate analysis of CD28null T cells and late allograft rejection

Clinical and immunological parameters, which were significant different between the late rejection and no late rejection group were used in a multivariate Cox regression analysis. Of the immunological parameters only the percentage and the absolute number of CD28null CD8 T cells remained significantly independent associated with late rejection (Table 4).

Kaplan-Meier curves for late rejection-free survival were made according to the tertile of percentage (Fig 1A) or absolute number (Fig 1B) of CD28null CD8 T cells.

In particular the recipients falling in the highest tertile of CD28null CD8 T cells had the lowest incidence of late allograft rejection.

In the highest tertile of percentages of CD28null CD8$^+$ T cells, the cumulative hazard of late rejection at 5 and 10 years was only 4% and 8% compared to 16% and 22% in the lowest tertile (p = 0.002).

This finding correlated with the risk for allograft loss because of late rejection, which was very low in recipients with a high percentage of CD28null CD8 T cells prior to transplantation (Fig 2, lowest versus highest tertile p-value = 0.001).

Multivariate regression analysis showed that both the tertile (HR 0.64 (95% CI 0.41–0.99), p = 0.04) and the % of CD28null CD8 T cells (HR 0.98 (95% CI 0.96–0.99), p = 0.02) were significantly associated with allograft loss because of late rejection.

**Table 4. Multivariate cox regression analysis for outcome late rejection after transplantation.**

| Multivariate model with clinical parameters and percentages of T cell subsets | | | |
|---|---|---|---|
|  | **Hazard ratio** | **95% confidence interval** | **p-value** |
| Previous transplantation | 2.19 | 1.12–4.34 | 0.024 |
| Dialysis before transplantation | 2.23 | 1.09–4.58 | 0.028 |
| Early acute rejection | 2.01 | 1.07–3.75 | 0.029 |
| % CD28null CD8 T cells | 0.98 | 0.96–0.99 | 0.007 |
| **Multivariate model with clinical parameters and absolute numbers of T cells subsets** | | | |
| Previous transplantation | 2.21 | 1.12–4.35 | 0.022 |
| Dialysis before transplantation | 2.16 | 1.05–4.36 | 0.035 |
| Early acute rejection | 2.12 | 1.13–3.95 | 0.019 |
| CD28null CD8 T cells/ul | 0.99 | 0.94–0.99 | 0.036 |

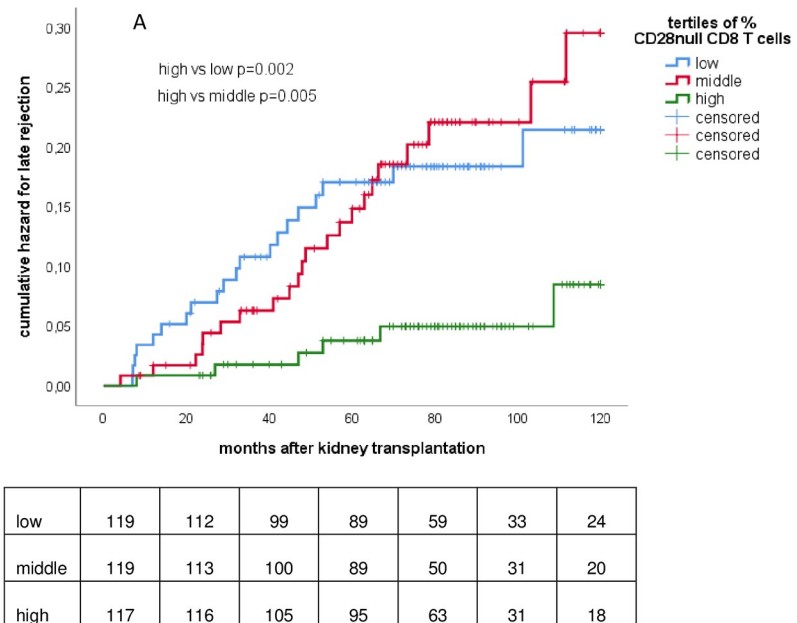

| low | 119 | 112 | 99 | 89 | 59 | 33 | 24 |
| middle | 119 | 113 | 100 | 89 | 50 | 31 | 20 |
| high | 117 | 116 | 105 | 95 | 63 | 31 | 18 |

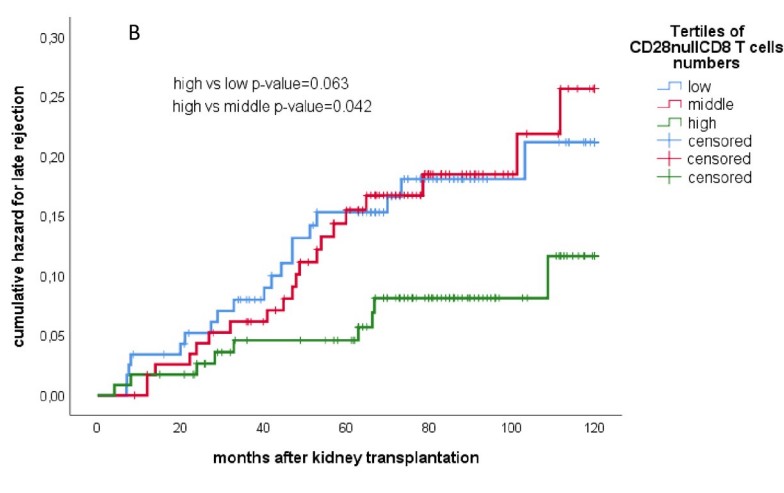

| low | 119 | 113 | 106 | 90 | 58 | 33 | 25 |
| middle | 119 | 114 | 106 | 89 | 51 | 31 | 21 |
| high | 117 | 113 | 98 | 94 | 63 | 31 | 16 |

**Fig 1. CD28 null CD8 T cells prior to transplantation and risk for late rejection.** Tertiles of CD28null CD8 T cells in percentage (A) and absolute number of cells (B) in relation to the cumulative hazard for late rejection after transplantation. P-values for difference between different strata were calculated by log-rank statistical analysis.

## Discussion

In this study the hypothesis was tested that increased immunological ageing at the time of renal transplantation carries a lower risk for late allograft rejection. The results show that specifically increased numbers of CD28null CD8 T cells are associated with a lower frequency of late rejection and graft loss because of late rejection.

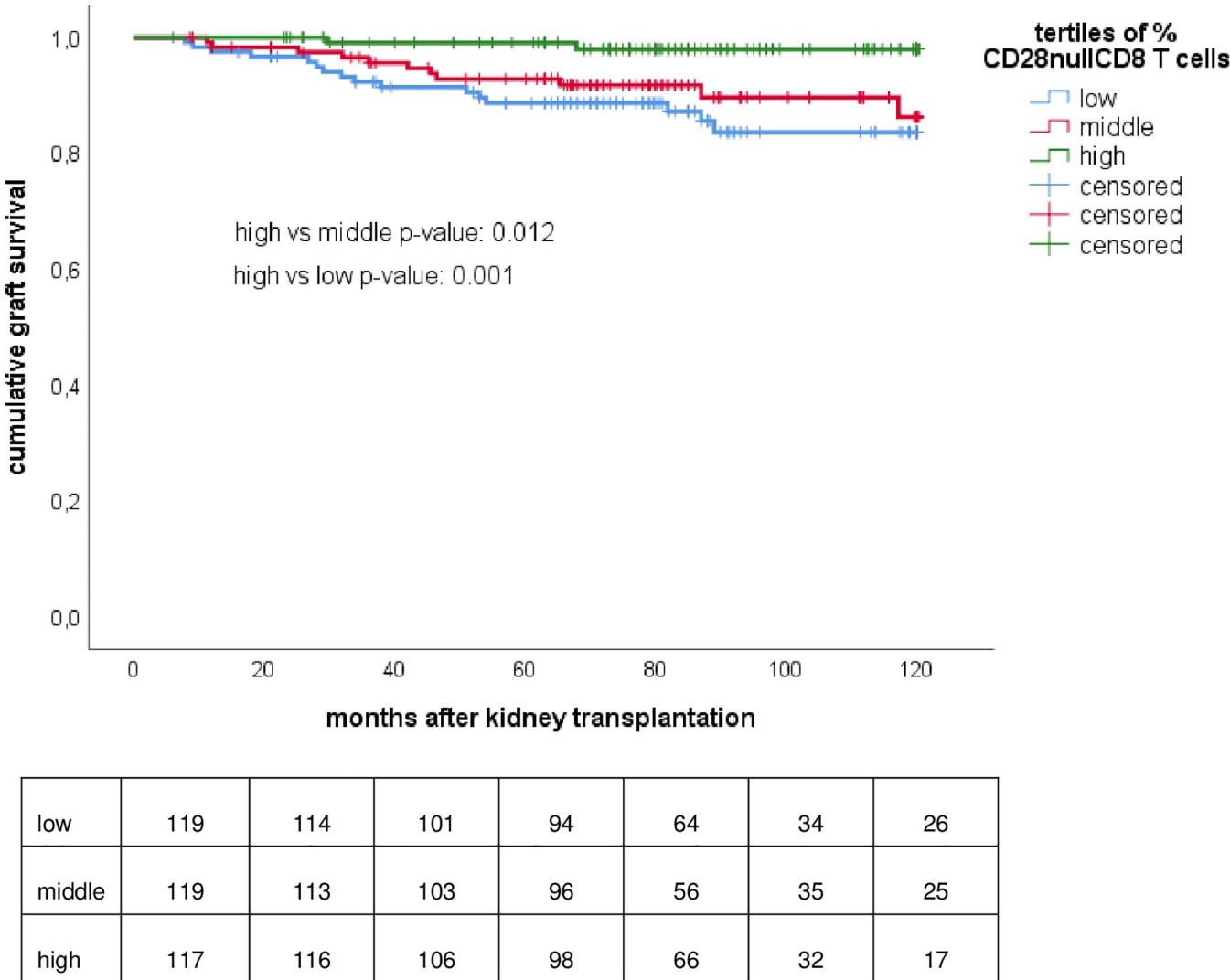

**Fig 2. CD28null CD8 T cells prior to transplantation and late rejection-related graft loss.** Tertiles of percentage of CD28null CD8 T cells and late rejection-related graft loss after transplantation. Differences between the Kaplan-Meier survival curves for the different strata were calculated by log-rank statistical analysis.

At present we have no obvious explanation for the inverse relation between an expanded pool of CD28null CD8 T cells and less late rejection. This relation appears to be independent of other markers of T cell ageing such as telomere length and thymus function, indicating a specific role of CD28null CD8 T cells in rejection. The data are in line with previous studies showing a similar relation of highly differentiated T cells with early acute rejection[10, 12]. Also, a lower frequency of CD28null CD8 T cells after alemtuzumab as induction therapy was associated with late rejection[28].

Alloreactive T cells can be identified in the CD28null T cell population and these cells are able to proliferate, but only in the presence of exogenous IL-15 or IL-21[16]. The CD28null T cells are predominantly present in the peripheral blood and only at low frequencies in the lymph nodes, consistent with the absence of the lymph node homing receptor CCR7 on these cells[15]. Given these findings it is most likely that CD28null T cells directly migrate into

tissues like the renal allograft. However, as more of these cells appear to protect against rejection their modus of operandi may actually be of a CD8 suppressor/regulatory T cell as identified by different studies[13]. Direct suppressor function of CD28null CD8 T cells on alloreactive T cell proliferation in vitro could not be shown by our group[10], but an indication of suppressor function for alloreactive CD4 T cells was shown by Trzonkowski et al[28]. Most in vitro studies on CD8 T cells suppressor function use some time of stimulation rather than a direct use of cells. Reasoning along the same line, some time period of stimulation in the tissues may be needed for full development of suppressor function.

Of note, this cell population may expand after CMV infection providing a potential confounder[29]. However, in this study the prevalence of a positive CMV serostatus was similar for both late rejection and non-rejecting groups.

The diagnosis of late rejection was made after a median time period of 44 months posttransplantation. Therefore, the dynamics of CD28null T cells after transplantation is of interest but there are only few studies that have studied this. Our group found very stable levels of these cells within the first year after kidney transplantation[30] and a similar stability of highly differentiated CD8 T cells was found at a prolonged time after transplantation[31]. In the latter study, a higher percentage of CD28null CD8 T cells was associated with increased risk for squamous cell cancer again indicative of a decrease in T cell immunity.

Of interest is the observation that after T cell depleting therapy the highly differentiated memory T cells are among the first to repopulate within the circulation[28]. This further underlines that CD8 CD28null T cells are not *per se* dangerous to the graft and may actually suppress T cell alloreactivity. Most likely the CD28null CD8 T cells are a heterogeneous cell population containing both effector cells and suppressor cells[13].

The strength of the current study is the number of recipients forming a relatively homogenous group largely transplanted with a kidney from a living donor and receiving the same immune suppressive drugs without initial depleting T cell therapy. In addition, the duration of follow-up and number of events is adequate for meaningful analysis. However, the potential weakness is that the present data do not allow for extrapolation of the findings to e.g. other immune suppressive drugs regimens and deceased donor kidney recipients. Also, there is no additional information on markers of senescence on CD28null T cells, which could have given more insight into the type of cell subset involved. Validation of the findings in a different cohort is of course essential but this will require a similar large number of recipients with immunophenotyping before transplantation and a follow-up of at least 5–10 years.

In conclusion, recipients with high numbers of CD28null CD8 T cells prior to transplantation are at a lower risk for late allograft rejection. Late allograft rejection contributes significantly to graft loss in the long term and prevention and treatment is considered an unmet need in transplantation. This is the first study showing that recipients may actually have an immunological profile protecting them from this threatening long-term complication. This knowledge may be used to guide tapering of immune suppressive medication and warrants further studies to elucidate the underlying mechanisms involved.

## Supporting information

**S1 File.**
(SAV)

**S1 Fig. Informed consent kidney donation by living donor.**
(TIF)

## Author Contributions

**Conceptualization:** Michiel G. H. Betjes.

**Data curation:** Nicolle H. R. Litjens.

**Formal analysis:** Michiel G. H. Betjes, Nicolle H. R. Litjens.

**Writing – original draft:** Michiel G. H. Betjes.

**Writing – review editing:** Nicolle H. R. Litjens.

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
