## [Decision Letter · Decision Letter 0]

14 Oct 2019

PONE-D-19-20193

High numbers of differentiated CD28null CD8+ T cells are associated with a lowered risk for late rejection and graft loss after kidney transplantation

PLOS ONE

Dear Dr. Betjes,

Thank you for submitting your manuscript to PLOS ONE. After careful consideration, we feel that it has merit but does not fully meet PLOS ONE’s publication criteria as it currently stands. Therefore, we invite you to submit a revised version of the manuscript that addresses the points raised during the review process.

We have now received reports from two referees of your manuscript, as agree with reviewers comments raised a few concerns about this study. After careful consideration, we invite you to submit a revised version of the manuscript.  

We would appreciate receiving your revised manuscript by Nov 28 2019 11:59PM. To enhance the reproducibility of your results, we recommend that if applicable you deposit your laboratory protocols in protocols.io, where a protocol can be assigned its own identifier (DOI) such that it can be cited independently in the future. For instructions see: http://journals.plos.org/plosone/s/submission-guidelines#loc-laboratory-protocols

We look forward to receiving your revised manuscript.

Kind regards,

Senthilnathan Palaniyandi, Ph.D

Academic Editor

PLOS ONE

Journal Requirements:

2.  We note that your study involved tissue/organ transplantation. Please provide the following information regarding tissue/organ donors for transplantation cases analyzed in your study.

1. Please provide the source(s) of the transplanted tissue/organs used in the study, including the institution name and a non-identifying description of the donor(s).

2. Please state in your response letter and ethics statement whether the transplant cases for this study involved any vulnerable populations; for example, tissue/organs from prisoners, subjects with reduced mental capacity due to illness or age, or minors.

- If a vulnerable population was used, please describe the population, justify the decision to use tissue/organ donations from this group, and clearly describe what measures were taken in the informed consent procedure to assure protection of the vulnerable group and avoid coercion.

- If a vulnerable population was not used, please state in your ethics statement, “None of the transplant donors was from a vulnerable population and all donors or next of kin provided written informed consent that was freely given.”

3. In the Methods, please provide detailed information about the procedure by which informed consent was obtained from organ/tissue donors or their next of kin. In addition, please provide a blank example of the form used to obtain consent from donors, and an English translation if the original is in a different language.

4. Please indicate whether the donors were previously registered as organ donors. If tissues/organs were obtained from deceased donors or cadavers, please provide details as to the donors’ cause(s) of death.

5. Please provide the participant recruitment dates and the period during which transplant procedures were done (as month and year)

6. Please discuss whether medical costs were covered or other cash payments were provided to the family of the donor. If so, please specify the value of this support (in local currency and equivalent to U.S. dollars).

Additional Editor Comments (if provided):

Reviewers' comments:

Reviewer's Responses to Questions

**Comments to the Author**

1. Is the manuscript technically sound, and do the data support the conclusions?

Reviewer #1: Yes

Reviewer #2: Yes

2. Has the statistical analysis been performed appropriately and rigorously? 

Reviewer #1: No

Reviewer #2: Yes

3. Have the authors made all data underlying the findings in their manuscript fully available?

Reviewer #1: Yes

Reviewer #2: Yes

4. Is the manuscript presented in an intelligible fashion and written in standard English?

Reviewer #1: Yes

Reviewer #2: Yes

5. Review Comments to the Author

Reviewer #1: In this study the authors characterized the T cell population prior to kidney transplantation in a cohort of 365 patients who received a kidney transplant form a living donor in the period 2007-2013 with the hypothesis that parameters indicative of an aged T-cell population associate with a lower risk for late rejection.

They found that the number and percentage of differentiated T cells were lower in patients with biopsy-proven late rejection (n. 50, diagnosed at a median of 44 months) and were independently and inveresely associated with rejection occurrence.

The study is well-conducted and the main strength is that a relatively large and homogeneous cohort of patients was analysed, with a relatevely long follow-up (median 70-80 months).

In my opinion, there are some points to be elucidated:

- were all the biopsies performed on clinical indication?

- was there a correlation between the T cell population before the transplantation and the type of rejection?

- The data about DSA pre-transplantation is missing; this would l be an important information in the evaluation of the immunological status of the recipients. Most of the rejection (39/50) were categorized as ABMR: were DSA searched at the time of the kidney biopsy? If so, it would be appropriate to evaluate the correlation beetwen DSA and T cell population before the transplantation, and the association of DSA with graft loss.

- in the Kaplan-Meier curves (Fig 1A, 1B and 2) the number of patients at risk should be reported

- In the Kaplan-Meier analysis for late rejection-related graft loss (Fig 2) the time considered should be the time after the rejection diagnosis, not the time after transplantation

- A multivariate analysis for graft loss would be needed

- page 11, line 176 should be "....had a significantly HIGHER percentage s of naive CD4 and CD8 T cells and LOWER percentages of the CD8 effector..."

- page 11, line 178 should be: "...corresponded with a significant LOWER percentage of CD28null CD8 T cells.."

Reviewer #2: This is a retrospective study in kidney transplant recipients to determine if there is an association between T cell telomere length and T cell subset analysis (CD28 and CD8) correlated with late (>6 months) acute rejection. Several questions arise:

1. The overall population is of relatively immunologic risk - unsensitized, living donors. Despite this, the "late rejection" group had a very high early acute rejection rate -33%. Additionally, nearly 20% of the entire population had a late acute rejection. Can you explain the very high rate of early and late acute rejection in this low risk group

2. No data is supplied regarding immunosuppressive levels at any time point post-tx. Given the very high rejection rates, this is an important variable that must be accounted for in your multi-variant analysis.

3. There is a very high rate of anti-body mediated rejection compared to cell mediated rejection. Please explain

4. Do you have data on development of donor specific antibody formation.

5. Nearly 40% did not receive any induction therapy. How was induction therapy chosen?

6. Can you expound on why telomere length had no influence on long-term outcomes?

6. PLOS authors have the option to publish the peer review history of their article (what does this mean?). If published, this will include your full peer review and any attached files.

Reviewer #1: No

Reviewer #2: No

---

## [Author Response · Author response to Decision Letter 0]

12 Nov 2019

PLOS ONE

Journal Requirements:

We have changed the manuscript accordingly.

2. We note that your study involved tissue/organ transplantation. Please provide the following information regarding tissue/organ donors for transplantation cases analyzed in your study.

1. Please provide the source(s) of the transplanted tissue/organs used in the study, including the institution name and a non-identifying description of the donor(s).

Eurotransplant in case of deceased donors kidneys and Erasmus MC for living donor kidney transplantations. This information is now provided in the revised manuscript.

2. Please state in your response letter and ethics statement whether the transplant cases for this study involved any vulnerable populations; for example, tissue/organs from prisoners, subjects with reduced mental capacity due to illness or age, or minors.

- If a vulnerable population was used, please describe the population, justify the decision to use tissue/organ donations from this group, and clearly describe what measures were taken in the informed consent procedure to assure protection of the vulnerable group and avoid coercion.

- If a vulnerable population was not used, please state in your ethics statement, “None of the transplant donors was from a vulnerable population and all donors or next of kin provided written informed consent that was freely given.”

A vulnerable population was not used and the required ethics statement is provided in the revised manuscript.

3. In the Methods, please provide detailed information about the procedure by which informed consent was obtained from organ/tissue donors or their next of kin. In addition, please provide a blank example of the form used to obtain consent from donors, and an English translation if the original is in a different language.

This procedure is now given in the Methods by referring to the legislation within the Eurotransplant region (Eurotransplant.org under legislation/eurotransplant) and the informed consent form for donors (translated in English) is included.

4. Please indicate whether the donors were previously registered as organ donors. If tissues/organs were obtained from deceased donors or cadavers, please provide details as to the donors’ cause(s) of death.

All deceased donor kidneys were allocated via Eurotransplant and donor registration was regulated per country according to national legislation. In our cohort, sixty percent of deceased donor kidneys were from deceased by cardiac death donors and 40% from deceased by brain death donors. This information is now given in the Methods section.

5. Please provide the participant recruitment dates and the period during which transplant procedures were done (as month and year)

The period during which the transplant procedures were performed is now provided in the revised manuscript.

6. Please discuss whether medical costs were covered or other cash payments were provided to the family of the donor. If so, please specify the value of this support (in local currency and equivalent to U.S. dollars).

All medical costs were covered by the medical insurance. This information is now provided in the revised manuscript.

Additional Editor Comments (if provided):

Reviewers' comments:

Reviewer's Responses to Questions

Comments to the Author

1. Is the manuscript technically sound, and do the data support the conclusions?

Reviewer #1: Yes

Reviewer #2: Yes

2. Has the statistical analysis been performed appropriately and rigorously? 

Reviewer #1: No

Reviewer #2: Yes

3. Have the authors made all data underlying the findings in their manuscript fully available?

Reviewer #1: Yes

Reviewer #2: Yes

4. Is the manuscript presented in an intelligible fashion and written in standard English?

Reviewer #1: Yes

Reviewer #2: Yes

5. Review Comments to the Author

Reviewer #1: In this study the authors characterized the T cell population prior to kidney transplantation in a cohort of 365 patients who received a kidney transplant form a living donor in the period 2007-2013 with the hypothesis that parameters indicative of an aged T-cell population associate with a lower risk for late rejection.

They found that the number and percentage of differentiated T cells were lower in patients with biopsy-proven late rejection (n. 50, diagnosed at a median of 44 months) and were independently and inveresely associated with rejection occurrence.

The study is well-conducted and the main strength is that a relatively large and homogeneous cohort of patients was analysed, with a relatevely long follow-up (median 70-80 months).

In my opinion, there are some points to be elucidated:

- were all the biopsies performed on clinical indication? Yes

- was there a correlation between the T cell population before the transplantation and the type of rejection? No

- The data about DSA pre-transplantation is missing; this would l be an important information in the evaluation of the immunological status of the recipients. Most of the rejection (39/50) were categorized as ABMR: were DSA searched at the time of the kidney biopsy? If so, it would be appropriate to evaluate the correlation between DSA and T cell population before the transplantation, and the association of DSA with graft loss.

We agree with this comment and data about DSA pre-transplantation and of most patients at time of biopsy would be of value. Unfortunately, this information is largely absent as DSA were not structurally assessed before and at time of rejection within the time period of this study. This is stated in the manuscript.

- in the Kaplan-Meier curves (Fig 1A, 1B and 2) the number of patients at risk should be reported

This is now provided in the KM curves in the revised manuscript.

- In the Kaplan-Meier analysis for late rejection-related graft loss (Fig 2) the time considered should be the time after the rejection diagnosis, not the time after transplantation

The research question is whether pre-transplant numbers of CD28null T cells are related to late rejection and rejection-related graft loss. Therefore, we considered time after transplantation to graft loss. If we would take time after rejection to graft loss then we would study the question whether the numbers of CD28null T cells are associated with progression to graft loss after rejection is diagnosed. This is obviously a different approach and would test the hypothesis that CD28null T cells can mitigate or worsen the clinical course of rejection. We did this analysis but could not find such an effect. 

- A multivariate analysis for graft loss would be needed

We now have included the results of the multivariate analysis for rejection-related graft loss in the results section of the revised manuscript. 

- page 11, line 176 should be "....had a significantly HIGHER percentage s of naive CD4 and CD8 T cells and LOWER percentages of the CD8 effector..."

- page 11, line 178 should be: "...corresponded with a significant LOWER percentage of CD28null CD8 T cells.."

These errors have now been corrected.

Reviewer #2: This is a retrospective study in kidney transplant recipients to determine if there is an association between T cell telomere length and T cell subset analysis (CD28 and CD8) correlated with late (>6 months) acute rejection. Several questions arise:

1. The overall population is of relatively immunologic risk - unsensitized, living donors. Despite this, the "late rejection" group had a very high early acute rejection rate -33%. Additionally, nearly 20% of the entire population had a late acute rejection. Can you explain the very high rate of early and late acute rejection in this low risk group

The group of recipients has a mixed profile for their immunological risk as panel reactive antibodies were present in some recipients (either at time of transplantation or historically) and re-transplantations were performed. The % of early rejection diagnosed within the whole population is not that different from data obtained from registries which is always higher than most clinical trials with study drugs including low-immunological risk patients only.

For instance, in a recent paper from the ANZDATA registry the % early AR was 21% in a large cohort of first kidney transplantations and a significant risk factor for late rejection (Clayton et al, J Am Soc Nephrol. 2019 Sep;30(9):1697-1707). 

The percentage of late rejections (which is not synonymous with acute rejection) is a cumulative percentage obtained at a rather long follow-up. Patients transplanted at our center have their regular visits at their referral medical center but are once-yearly seen by a transplantation-nephrologist in our center. When a steady decline in renal allograft function is observed we perform a diagnostic kidney biopsy to rule out late (chronic) rejection as a cause. In this way we try to diagnose all cases of late rejection which yields a much higher number of late rejections than most studies with a low % of diagnostic biopsies late after transplantation (e.g. Naesens et al, Transplantation. 2014 Aug 27;98(4):427-35)

2. No data is supplied regarding immunosuppressive levels at any time point post-tx. Given the very high rejection rates, this is an important variable that must be accounted for in your multi-variant analysis.

This is a very difficult question to address. The large majority of patients were given tacrolimus in combination with MMF and prednisone within the first 6 months after transplantation with through levels for tacrolimus by protocol. This protocol is now mentioned within the revised manuscript. Thereafter the data on through levels are incomplete as most patients were seen in their referral nephrology center and combinations of immune suppressive drugs could be changed. In addition, emerging data strongly suggest that the degree of intra-individual variability is a much stronger predictor of graft survival than through levels only. In a recent paper we did analyze the through levels and IPV in patients with late humoral rejection in great detail (Sablik et al, Transplant Int 31: 900-908, 2018 ) and found that the average tacrolimus through level was 6 which is above the general recommended through level of 5 ug/ml.

3. There is a very high rate of anti-body mediated rejection compared to cell mediated rejection. Please explain

Late rejection is frequently caused by anti-body mediated rejection and in the years after transplantation the incidence of cell mediated rejection becomes (very) low. 

4. Do you have data on development of donor specific antibody formation.

Unfortunately, we have only data on DSA in the minority of patients before and after kidney transplantation as DSA assessment was not routinely performed. This is stated in the manuscript.

5. Nearly 40% did not receive any induction therapy. How was induction therapy chosen?

Within the study period, the use of anti-CD25 antibody was not yet the standard induction therapy. Induction with anti-CD25 antibody or a T cell depleting agent was done when the risk for acute rejection was thought to be high (e.g. high %PRA, high number of MM, frequent rejections in the past or with a repeated mismatch with a previous transplant).

6. Can you expound on why telomere length had no influence on long-term outcomes?

 Telomere length is assessed on the whole populations of CD4 and CD8 T cells. It has a significant but variable relation with the recipient age and therefore is a “bulk” marker of immunological ageing. The association of an expanded CD28null T cell population and less rejection seems to be more cell specific and, although speculative, may indicate a role of e.g. suppressor CD8 T cells. This has been stated in the discussion of the manuscript.

While revising your submission, please upload your figure files to the Preflight Analysis and Conversion Engine (PACE) digital diagnostic tool, https://pacev2.apexcovantage.com/. PACE helps ensure that figures meet PLOS requirements. To use PACE, you must first register as a user. Registration is free. Then, login and navigate to the UPLOAD tab, where you will find

This has been performed.

---

## [Decision Letter · Decision Letter 1]

27 Nov 2019

PONE-D-19-20193R1

High numbers of differentiated CD28null CD8+ T cells are associated with a lowered risk for late rejection and graft loss after kidney transplantation

PLOS ONE

Dear Dr. Betjes,

Thank you for submitting your manuscript to PLOS ONE. After careful consideration, we feel that it has merit but does not fully meet PLOS ONE’s publication criteria as it currently stands. Therefore, we invite you to submit a revised version of the manuscript that addresses the points raised during the review process.

We have now received reports from referees of your revised manuscript, there are few areas need to be revised to make this manuscript as complete. 

We would appreciate receiving your revised manuscript by Jan 11 2020 11:59PM. To enhance the reproducibility of your results, we recommend that if applicable you deposit your laboratory protocols in protocols.io, where a protocol can be assigned its own identifier (DOI) such that it can be cited independently in the future. For instructions see: http://journals.plos.org/plosone/s/submission-guidelines#loc-laboratory-protocols

We look forward to receiving your revised manuscript.

Kind regards,

Senthilnathan Palaniyandi, Ph.D

Academic Editor

PLOS ONE

Reviewers' comments:

Reviewer's Responses to Questions

**Comments to the Author**

1. If the authors have adequately addressed your comments raised in a previous round of review and you feel that this manuscript is now acceptable for publication, you may indicate that here to bypass the “Comments to the Author” section, enter your conflict of interest statement in the “Confidential to Editor” section, and submit your "Accept" recommendation.

Reviewer #1: All comments have been addressed

Reviewer #2: (No Response)

2. Is the manuscript technically sound, and do the data support the conclusions?

Reviewer #1: Yes

Reviewer #2: Partly

3. Has the statistical analysis been performed appropriately and rigorously? 

Reviewer #1: Yes

Reviewer #2: Yes

4. Have the authors made all data underlying the findings in their manuscript fully available?

Reviewer #1: Yes

Reviewer #2: Yes

5. Is the manuscript presented in an intelligible fashion and written in standard English?

Reviewer #1: (No Response)

Reviewer #2: Yes

6. Review Comments to the Author

Reviewer #1: (No Response)

Reviewer #2: This is a revised manuscript of a retrospective study in kidney transplant recipients to determine if there is an association between T cell telomere length and T cell subset analysis (CD 28 and CD 8) correlated with late (>6 months) acute rejection.

I appreciate your response to both reviewers. Several questions still exist:

1. It is still not clear why there is a high rate of acute rejection is a very low immunological risk population. The overwhelming majority of recipients were unsensitized, living donor recipients. Acute rejection rates of 19.5% at 6 months in this group is still far higher than data from the Europenan EKiTE registry (Lorent, M., Foucher, Y., Kerleau, K. et al. The EKiTE network (epidemiology in kidney transplantation - a European validated database): an initiative epidemiological and translational European collaborative research. BMC Nephrol 20, 365 (2019) doi:10.1186/s12882-019-1522-8) showing a 79% 2 year rejection free in a population consisting of 81% deceased donors with a large portion that were sensitized or repeat transplants.

2. How was the diagnosis of "late" acute rejection made? You mention that late rejections are not synonymous with acute rejection

3. You note that late rejections are most often antibody mediated but you have limited DSA data. Was the diagnosis of AMR made exclusively on the basis of C4D?

4. The lack of immunosuppressive drug levels is a significant shortcoming

5. The selection of induction agents remains unanswered. The choice of anti-DC25 or ATG is a very different class of agents vs IL2-RA. The immunologic effect can be very different on your T cell subsets. Have you analyzed these groups separately to determine if there is any difference in outcomes.

7. PLOS authors have the option to publish the peer review history of their article (what does this mean?). If published, this will include your full peer review and any attached files.

Reviewer #1: No

Reviewer #2: No

---

## [Author Response · Author response to Decision Letter 1]

3 Jan 2020

Dear editor,

The comments of reviewer #2 are adressed in a point-by-point fashion as shown below.

We hope the manuscript is now suitable for publication.

Sincerely,

Dr Michiel Betjes

I appreciate your response to both reviewers. Several questions still exist:

1. It is still not clear why there is a high rate of acute rejection is a very low immunological risk population. The overwhelming majority of recipients were unsensitized, living donor recipients. Acute rejection rates of 19.5% at 6 months in this group is still far higher than data from the Europenan EKiTE registry (Lorent, M., Foucher, Y., Kerleau, K. et al. The EKiTE network (epidemiology in kidney transplantation - a European validated database): an initiative epidemiological and translational European collaborative research. BMC Nephrol 20, 365 (2019) doi:10.1186/s12882-019-1522-8) showing a 79% 2 year rejection free in a population consisting of 81% deceased donors with a large portion that were sensitized or repeat transplants.

We agree with the reviewer that in our patient population a relatively high rate of acute rejection is observed. The reasons are difficult to identify as we use a triple immune suppressive regimen including prednison/MMF/tacrolimus with adequate through levels defined by protocol (see M&M). Comparing patient populations is always difficult with potential unrecognized bias. For instance, the EKiTE data show T cell depletion as induction therapy in 37% of recipients while we rarely give this kind of induction therapy in the Netherlands. In other countries like the USA this percentage can be as high as 70% in some centres. On the other hand, publications from Japan and Korea show a very low AR percentage without T cell depletion and a standard IS regimen.

Virtually all our episodes of rejection are biopsy proven. In other studies, underreporting of rejection may occur as no biopsy was done before anti-rejection therapy. Our kidney biopsies are scored by a board-certified renal pathologist but we cannot rule out the possibility that our pathologists are leaning more to a diagnosis of rejection (for instance in cases of borderline acute rejection) than their colleagues in other centres.

In summary, we cannot satisfactorily address this comment but can only speculate on possible explanations. Of note, despite a relatively high AR rate the one-year graft and patient survival of our living donor kidney transplant program is excellent (>98%).

2. How was the diagnosis of "late" acute rejection made? You mention that late rejections are not synonymous with acute rejection

Late rejections are by definition occurring 6 months (admittedly our definition but a commonly used definition) after kidney transplantation. A rapid decline in renal function with a biopsy showing little chronicity but predominantly an influx of immune cells occurring many months after transplantation is compatible with an acute rejection and can be observed when e.g. patients are significantly under immune supressed. However, in most cases there is a progressive decrease in eGFR loss over a period of weeks to months with a biopsy showing rejection with signs of chronicity.

3. You note that late rejections are most often antibody mediated but you have limited DSA data. Was the diagnosis of AMR made exclusively on the basis of C4D?

All our renal biopsies were evaluated by an experienced renal pathologist who scores >150 renal transplant biopsies each year in our centre. The diagnosis of AMR was based on using the Banff criteria but C4D staining can be absent. However, a renal biopsy with all histological hallmarks of AMR was diagnosed as AMR after discussing the case with the nephrologists at our weekly renal pathology meeting. The Banff 2015 allows for this by having a category of suspicious ABMR when DSA and/or C4d are negative while all other criteria are met.

4. The lack of immunosuppressive drug levels is a significant shortcoming

We have a protocol in place for through levels of MMF and tacrolimus and at our centre we adhere quite strictly to this protocol as outlined in the M&M section. However, patients are eventually referred to their own nephrologist for follow-up and we cannot account for this period. Single through levels are not that reliable and there is a growing body of evidence that high intrapatient variability of tacrolimus through levels is actually more predictive of future rejection and graft loss. Unfortunately, we do not have these data but also cannot think of a reason why IPV could give a structural bias in our data (that is consistently more IPV in a subgroup of patients with high or low numbers of CD28null T cells).

5. The selection of induction agents remains unanswered. The choice of anti-DC25 or ATG is a very different class of agents vs IL2-RA. The immunologic effect can be very different on your T cell subsets. Have you analyzed these groups separately to determine if there is any difference in outcomes.

We are not sure what to answer. A very low number of patients received ATG induction and this is to small a group for meaningful statistical analysis. Anti-CD25 is synonymous with IL-2-receptor antagonist.

Within a multivariate analysis, the use of basiliximab (anti-CD25) was not a significant variable associated with late rejection. All these date are shown in table 1.

Within the discussion we recognize that the results obtained are possibly not generalizable when other IS drug regimes are used, notably induction with T cell depleting agents.

---

## [Editor Report · Decision Letter 2]

8 Jan 2020

High numbers of differentiated CD28null CD8+ T cells are associated with a lowered risk for late rejection and graft loss after kidney transplantation

PONE-D-19-20193R2

Dear Dr. Betjes,

We are pleased to inform you that your manuscript has been judged scientifically suitable for publication and will be formally accepted for publication once it complies with all outstanding technical requirements.

With kind regards,

Senthilnathan Palaniyandi, Ph.D

Academic Editor

PLOS ONE
---

## [Editor Report · Acceptance letter]

15 Jan 2020

PONE-D-19-20193R2 

High numbers of differentiated CD28null CD8+ T cells are associated with a lowered risk for late rejection and graft loss after kidney transplantation 

Dear Dr. Betjes:

I am pleased to inform you that your manuscript has been deemed suitable for publication in PLOS ONE. Congratulations! Your manuscript is now with our production department. 

With kind regards,

on behalf of

Dr. Senthilnathan Palaniyandi 

Academic Editor

PLOS ONE